# Tape-Shaped, Multiscale, and Continuous-Readable Fiducial Marker for Indoor Navigation and Localization Systems

**DOI:** 10.3390/s24144605

**Published:** 2024-07-16

**Authors:** Benedito S. R. Neto, Tiago D. O. Araújo, Bianchi S. Meiguins, Carlos G. R. Santos

**Affiliations:** 1Departamento de Ensino, Pesquisa, Pos-Graduação, Inovação e Extensão, Campus Cametá, Instituto Federal do Pará (IFPA), Cametá 68400-000, Pará, Brazil; 2Escola Superior Aveiro Norte (ESAN), Universidade de Aveiro, 3810-193 Aveiro, Portugal; tiagodavi70@gmail.com; 3Programa de Pós-Graduação em Ciência da Computação (PPGCC), Universidade Federal do Pará (UFPA), Belém 66075-110, Pará, Brazil; bianchi@ufpa.br (B.S.M.); gustavo.cbcc@gmail.com (C.G.R.S.)

**Keywords:** fiducial marker, indoor localization, smartphone, tape-shaped marker

## Abstract

The present study proposes a fiducial marker for location systems that uses computer vision. The marker employs a set of tape-shaped markers that facilitate their positioning in the environment, allowing continuous reading to cover the entire perimeter of the environment and making it possible to minimize interruptions in the location service. Because the marker is present throughout the perimeter of the environment, it presents hierarchical coding patterns that allow it to be robust against multiple detection scales. We implemented an application to help the user generate the markers with a floor plan image. We conducted two types of tests, one in a 3D simulation environment and one in a real-life environment with a smartphone. The tests made it possible to measure the performance of the tape-shaped marker with readings at multiple distances compared to ArUco, QRCode, and STag with detections at distances of 10 to 0.5 m. The localization tests in the 3D environment analyzed the time of marker detection during the journey from one room to another in positioning conditions (A) with the markers positioned at the baseboard of the wall, (B) with the markers positioned at camera height, and (C) with the marker positioned on the floor. The localization tests in real conditions allowed us to measure the time of detections in favorable conditions of detections, demonstrating that the tape-shaped-marker-detection algorithm is not yet robust against blurring but is robust against lighting variations, difficult angle displays, and partial occlusions. In both test environments, the marker allowed for detection at multiple scales, confirming its functionality.

## 1. Introduction

An indoor localization system is a technology that continuously estimates the position of objects or people in an internal environment [1]. Some technologies applied in positioning systems use wireless sensors such as Bluetooth [2], ZigBee [3], Wi-Fi [4], as well as hybrid approaches such as sensor fusion [5] that improve the accuracy and continuity of the location service. With the advancement of mobile computing [6], it is possible to identify the position of mobile/fixed devices including smartphones, drones, watches, beacons, and vehicles that can be used in different services, including navigation, tracking, and monitoring, which can be employed for localization both indoors and outdoors [7].

The positioning approach that uses cameras and employs computer vision techniques [8] can be considered low cost to implement when a simple smartphone camera and visual marks are sufficient to run the localization system, providing a reliable service [9].

Li et al. [10] report two categories of markers that can be used in localization systems, natural and artificial, as shown in Figure 1. Artificial markers generally face challenges with varying lighting and detection across different scales. They are detected quickly and accurately, as they are designed based on known coding rules. Using natural markers avoids changes to the environment’s internal infrastructure, allowing for the exploration of static physical objects or scenes from the internal environment, such as doors and windows.

Applications that employ fiducial markers include tags, a detection algorithm, and a coding system [11]. The detection process can be performed through algorithms based on traditional image-processing techniques such as edge detection, blob detection, image binary, or machine-learning-based detection [12].

**Figure 1 sensors-24-04605-f001:**
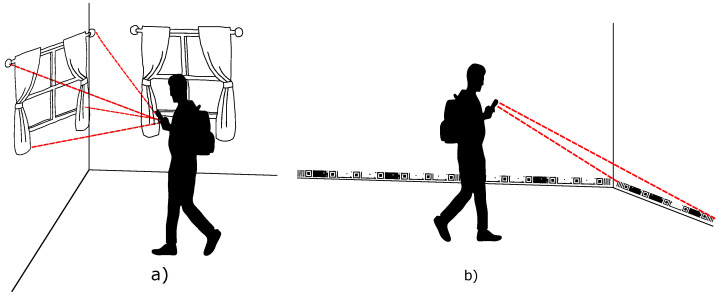
Illustration of indoor location system using smartphone. (**a**) With natural markers [10], (**b**) with a tape-shaped marker around the entire perimeter [13].

In an indoor localization system, fiducial markers are spread in the environment [14], generating gaps between markers that can cause the discontinuity of reading and obstruction of the localization service for some time so that the user would have to reposition the camera at another marker to restore the location service. From this perspective, the hypothesis arises of positioning the markers linearly with minimum distances between them to allow for a continuous reading of the environment so that there is always a marker in the image-capture scene and consequently minimizing the discontinuity of the location service, as exemplified in Figure 1b.

This work is an expanded form of the prototype [13] published at a conference that differs by (a) addressing a complete state of the art on fiducial markers applied in localization systems, (b) presenting a new component in the design of the marker for validating the information encoded in the marker, (c) presenting the web module for generating the marker in the form of a tape from a bitmap image of the floor plan of a property, (d) presenting a simulation of tests in a 3D environment, (e) presenting the development of the mobile application for reading the marker, and (f) presenting the experimental tests of the localization system using the tape-shaped marker in a natural environment.

In this context, the present study aims to present a fiducial marker for an indoor location system based on computer vision technology that uses a fiducial marker in the form of a tape with multiscale detections using a smartphone. This enables the location service to always remain operational, even with variations in the distances between the camera and the target.

Consequently, this study will help researchers choose technology for a low-cost indoor location system to be applied in the most varied human mobility environments and even robotics.

In addition to this introductory section, Section 2 covers related work. Section 3 presents an overview of the fiducial marker system. Section 4 reports the experimental tests and results of the indoor localization process using tape markers. Section 5 presents the discussion, and Section 6 presents the conclusions and future works.

## 2. Related Works

The state of the art in this field presents a variety of markers with different characteristics, such as color, format, and robustness against variations in application environment conditions.

Kalaitzakis et al. [15] address a literature review on characteristics such as the shape, color, and coding of 21 fiducial markers used in localization systems, in addition to comparing the performance of markers and ARTag [11], AprilTag [16], ArUco [17], and STag [18] addressing accuracy, detection rate, and computational cost in various testing scenarios with shadowing and blurring noise.

Wu et al. [19] present indoor location systems based on camera images, classified into systems with a known environment and systems in which the environment is not previously known. For systems with an unknown environment, different forms of SLAM are emphasized, divided into four categories: geometric SLAM, learning SLAM, topology SLAM, and the SLAM marker, such as sSLAM [20], which employs fiducial markers with circular and square shapes.

In addition to these works, Table 1 summarizes the main characteristics of markers available in the literature, such as format, color, coding, and the algorithm used in the detection process. Jurado-Rodríguez et al. [21] designed Jumarker to have an attractive and customizable appearance for location systems and can have a cubic shape with multiple faces. The coding of this marker consists of the identification pattern and the redundancy check pattern, which allows for the differentiation of markers even with partial occlusion.

Toyoura et al. [22] designed a polychromatic marker for localization systems called a monospectrum marker. It uses a two-dimensional sinusoidal intensity pattern with several colors to change the brightness at a single low frequency in the color spectrum. This low-frequency component is little affected by blurring. Thus, the regions in the images corresponding to the markers also have a single low-frequency signal, allowing them to be robust against lighting variation and blurring.

Bencina et al. [23] present the ReacTIVision fiducial marker that contains a topological structure that allows for the coding of information using any regular or irregular formats, as the detection process employs topological fiducial recognition introduced by Costanza and Robinson in the D-touch marker [24], which generates a graph of adjacency regions with values derived from a binary image of the scene through the segmentation process, allowing you to accurately calculate location and orientation without resorting to additional information extracted by image-processing techniques, such as corner or edge detection.

Wang et al. [25] present a fiducial marker intended for localization systems called the HArCo marker that presents the hierarchical structure based on ArUco [17] as it is possible to estimate the positioning within a much greater range if designed correctly. Even when the marker is partially occluded, its child markers are still available for pose estimation in the localization process.

Benligiray, Topal, and Akinlar [18] present the marker STag, which is highly robust in difficult viewing angle conditions for localization systems, as the performance of STag is significantly better than 80° compared to ARToolKitPlus [26].

In the FourrierTag published by Sattar et al. [27], a grayscale image is sufficient for detection as bits are encoded in the frequency domain, with successive lower-order bits using successively higher frequencies. Although the marker is perfectly circular, shortening the perspective will cause some degree of shape distortion. However, the shapes formed by the markers will always exhibit symmetry to some degree. Furthermore, the algorithm needs to extract a radius that extends from the center to the edge to decode the information encoded in the marker. Therefore, it is essential to find the center of the marker accurately. Otherwise, lightning extraction will not be possible, thus complicating the decoding task.

Schweiger et al. [28] employ SIFT and SURF descriptors that allow for the generation of signatures with distinct dark and light pixel values. However, the entries corresponding to the sum of absolute gradient values are identical. Furthermore, this variability of pixel values allows these markers to be detected at different angles. The other markers have binary colors (black and white) that allow for the encoding of bits 0 and 1.

**Table 1 sensors-24-04605-t001:** The main morphological and behavioral characteristics identified in fiducial markers in alphabetical order are used in localization systems.

Markers	Shape	Color	Encoding	Algorithm
AprilTag [16]	Square	Monochrome	Code	Geometric calculations
AprilTag2 [29]	Square	Monochrome	Code	Geometric calculations
AprilTag3 [30]	Square	Monochrome	Code	Geometric calculations
AprilTags 3D [31]	Square	Monochrome	Code	Geometric calculations
ArUco [17]	Square	Monochrome	Code	Geometric calculations
BlurTags [32]	Square	Monochrome	Code	Geometric calculations
BullsEye [33]	Circle	Monochrome	Code	Geometric calculations
Cantag [34]	Circle	Monochrome	Code	Geometric calculations
CCTag [35]	Circle	Monochrome	Code	Geometric calculations
Chilitags [36]	Square	Monochrome	Code	Geometric calculations
ChromaTag [37]	Square	Multicolor	Code	Geometric calculations
Claus and Fitzgibbon [38]	Square	Monochrome	Glyph	Trained
Color marker-based [39]	Triangulated	Multicolor	Code	Geometric calculations
Concentric contrasting circle [40]	Circle	Monochrome	Code	Geometric calculations
Concentric ring fiducial [41]	Circle	Monochrome	Code	Geometric calculations
CoP-Tag [42]	Square	Monochrome	Code	Geometric calculations
CyberCode [43]	Square	Monochrome	Code	Geometric calculations
DeepTag [12]	Square	Multicolor	Glyph	Trained
E2ETag [44]	Square	Monochrome	Code	Trained
Farkas et al. [45]	Square	Multicolor	Code	Geometric calculations
FourierTag [27]	Circle	Monochrome	Code	Geometric calculations
Fractal Marker [46]	Square	Monochrome	Code	Geometric calculations
HArCo marker [25]	Square	Monochrome	Code	Geometric calculations
ICL [47]	Square	Monochrome	Code	Region adjacency
Jumarker [21]	Cube	Multicolor	Code	Geometric calculations
LFTag [48]	Square	Multicolor	Code	Region adjacency
Markers with alphabet [49]	Cube	Monochrome	Glyph	Trained
Monospectrum marker [22]	Square	Multicolor	Code	Geometric calculations
Order Type Tags [50]	Square	Monochrome	Code	Geometric calculations
Pi-Tag [51]	Square	Monochrome	Code	Geometric calculations
PRASAD et al. [52]	Square	Monochrome	Code	Geometric calculations
ReacTIVision [23]	Undefined	Monochrome	Code	Region adjacency
RuneTag [53]	Circle	Monochrome	Code	Geometric calculations
Seedmarkers [54]	Undefined	Monochrome	Code	Region adjacency
SIFT [28]	Square	Monochrome	Code	Geometric calculations
sSLAM [20]	Square	Monochrome	Code	Geometric calculations
STag [18]	Square	Monochrome	Code	Geometric calculations
Standard Pattern [55]	Rectangle	Monochrome	Code	Geometric calculations
SURF [28]	Square	Monochrome	Code	Geometric calculations
SVMS [56]	Square	Monochrome	Code	Geometric calculations
Tcross [57]	Square	Multicolor	Code	Trained
Topotag [58]	Square	Monochrome	Code	Region adjacency
TRIP [59]	Circle	Monochrome	Code	Geometric calculations
WhyCode [60]	Circle	Monochrome	Code	Geometric calculations
X-tag [61]	Square	Monochrome	Code	Geometric calculations

ChromaTag [37] has a robust detection algorithm for lighting variations because it uses differences in chrominance and luminance throughout detection and localization. Liu et al. [39] presented another work with a detection algorithm robust against lighting variations with color marker-based markers. Their algorithm uses an adaptive threshold before extracting colors from the marker. They showed that using a fixed threshold method instead would result in a loss of color and a lack of robustness against lighting intensity.

The LFTag [48] is built to resolve rotational ambiguity (when the same tag delivers different IDs at different angles), which, combined with the robust geometric and topological rejection of false positives, allows all bits of the tag to be data. The key points present in the marker resolve the rotational ambiguity. The remaining marker regions are called “data regions”, and each encodes two bits in their relative location.

Tcross [57] employs the convolutional neural network YOLOv3 [62], which requires only colored images and a file with bounding box information. Before generating and entering the neural network, the images are resized to a resolution of 416 × 416 pixels with a dataset divided into 385 training images, 110 validation images, and 55 test images. This marker training takes 50 epochs, with the best results obtained in epoch 37. With these detection characteristics, Tcross [57] is robust against partial occlusion, and blurring can be detected at different angles and with variations in lighting. The disadvantage of this training-based detection approach is that it requires powerful hardware, time, and expertise for training.

Calvet et al. [35] present a robust, highly accurate fiducial marker called CCTAG, intended for localization systems, whose markers consist of monochromatic concentric rings that allow for robust and precise localization in images under very challenging conditions such as low lighting, blurring, and partial occlusion. The CCTag [35] allows it to be used in navigation applications resistant to partial occlusions, at varying distances and viewing angles, and with rapid camera movements, allowing the navigation system to be reliable and robust in applications that use robots or drones.

Bergamasco et al. [63] present RuneTag, a fiducial marker that explores the projective properties of a circular set of points in fixed angular positions. This allows the marker to be detected even with up to 70% of the features occluded, with blurring, and in lighting variate conditions.

Zhang et al. [12] present the DeepTag, intended for localization systems, which differ in their robustness because they are detected with sophisticated machine learning techniques such as artificial convolutional neural networks (CNNs) that require an abundance of images for training, including colored images with different variations in luminosity. This way, the neural network will learn to identify these markers so that they can be detected in various lighting conditions in localization systems.

The state of the art points to a diversity of fiducial markers for indoor location applications, although they do not address a marker that can be fixed around the entire perimeter of the environment, continuously visible throughout the user’s walk within the environment, and continuously tracked even with partial occlusions.

Furthermore, none of the markers mentioned in Table 1 explore robustness against multiple distances with multiple detection points linearly. This work allows for detection in a linear and horizontal way, with reading at several detection points sequentially, identifying parts of the marker, and horizontally, identifying the marker in its entirety, at short and long distances. Furthermore, our marker has more than 1 million different coding combinations; it has coding based on black and white ink levels with positive or negative values. It presents a simplified interface for generating markers for large areas, allowing for the generation of the marker from drawings on the floor plan of room grids, and can also be enlarged or reduced according to the need for use with a 25:7 aspect ratio for visual adaptation to the environment, which can be fixed in places such as skirting boards or at the top of the wall, reducing visual pollution, as shown in Figure 1b.

## 3. System Overview

This section presents an overview of the fiducial markers system with the web module for generating the tape and configuration files. The mobile module uses the tape-reading application on multiple scales to locate the internal environment, as illustrated in Figure 2.

The following subsections describe the fiducial marker proposed with the tape-shaped marker design, the coding process, the tape marker generation and mapping process from a floor plan, the multiscale detection process, and the tape-reading demonstration.

### 3.1. Marker Design

This subsection presents the morphological characteristics of the marker in the form of a tape that aims to map an internal environment linearly and continues along the path the user takes, even with the partial occlusion of some segment of the tape.

The proposed marker comprises a sequence of Code Markers (CMs) that can be read individually, especially when the camera is close to the target. A CM’s sequence generates a coding string called tape markers (TMs). This sequence can be read, especially when the camera is far from the marker. The TM reading algorithm is similar to the Standard Pattern [55] algorithm, which has parallel vertical bars at its ends.

The marker structure was designed to cover the room’s perimeter in areas such as skirting boards or ceiling edges, allowing the user to move around all spaces without losing visual contact with the tape.

The monochromatic marker facilitates the detection and extraction of information, including in variable lighting and low-resolution conditions. Figure 3 illustrates the Finder Pattern inspired by QRCode [64], as its geometric shape is easy to detect. The Alignment Patterns and coding region are similar to CyberCode [43] to adjust the ideal focal distance for recognizing and extracting coded information.

The CM can be positive, as shown in Figure 3a, when the background is white and the bits are black. Another possibility is that the CM presents coding for negative values, as seen in Figure 3b when the background is black and the bits are white.

The CM presents coding elements for positive values, as illustrated in Figure 3. Furthermore, each CM contains blocks arranged in a regular rectangular matrix containing the following components:Finder Patterns allow for marker location detection and positioning. They are located at the marker ends (left and right) of each symbol; they have 7 × 7 dark blocks, 5 × 5 light blocks, and a dark block in the center of 3 × 3.Quiet Zone is an area that does not contain data and is used to ensure that the surrounding markup does not disorient the marker code data made up of white blocks around the Finder Patterns, Alignment Patterns, and Region of Codification.Alignment Patterns are the black squares that form an ’L’ in a CM, containing a black square in the upper right corner of the region, providing marker orientation in scenes.The encoding region contains black and white blocks corresponding to the coded information embedded in the marker.Checksum contains eight black or white blocks corresponding to the validation bits of the information embedded in the marker.

The composition of a TM is shown in Figure 3c, which presents six segments of CMs, four with positive values and two with negative values, in addition to four vertical parallel bars that will always define the beginning and end of the TM.

The CM quantity varies within the tape according to the desired number of bits and the readability of the tape’s physical length because the more bits (the entire CM is a bit of the TM) are present in a TM, the longer that tape segment becomes. Thus, the longer the segment, the further the user needs to be from the target to frame an entire segment.

On the other hand, if too few bits are used, there will not be enough code to cover the entire space, depending on the available length. For example, using only three CMs per TM, there were only eight different codes to map the space.

### 3.2. Marker Encoding

A CM’s numerical codes must use the smallest possible variation in ink (bits 1 white color) for close numbers so that, when viewed from a long distance, they can be recognized as a single bit, with little ink for the positive and a lot of ink for the negative.

Therefore, the standard binary encoding, the Weighted Binary Code (WBC), hinders this feature as the number of bits can vary significantly between nearby numbers. For example, the binary version of 128 has one 1 and seven 0s, while the number 127 has seven 1s and one 0, considering a binary code of 8 bits.

Thus, we use an alternative binary encoding in which the first numbers are all combinations of one bit 1 and other bits 0, followed by all combinations of two-digit 1 s and the remaining digit 0 s, and so on. Table 2 illustrates a binary encoding with four bits using both encodings. The table shows that in the proposed encoding, the number of bits 1 grows in a stable and orderly manner, and the number of bits grows or changes in an unordered manner for the same sequence in the WBC encoding.

The proposed coding using four digits allows the use of numbers from 0 to 4 for less ink and from 11 to 15 for more ink (these ranges grow along with the length of the code). Furthermore, Table 2 shows some problems when using the numbers 3 and 12 with WBC encoding that can cause a duplication of information with too little or too much ink when the marker is far from the camera to read the TM, which needs to identify the positive CM low ink represented by bit 1 or the CM with little ink represented by bit 0.

This way, it is possible to create a sequence of codes that uses the smallest number of bits 1 possible, efficiently controlling the amount of ink in the available area of the marker. This characteristic is relevant because a TM comprises CMs that can be positive (majority white) or negative (majority black) and will be differentiated depending on the amount of ink used in these CMs.

The conversion from decimal to the proposed encoding starts by checking how many bit 1s are needed to represent the decimal value using the combinatorial formula. Then, all the necessary 1s are on the right, and the remaining 0s are on the left, forming an initial code. The decimal value of the initial code is the sum of all combinations with fewer bits. According to Algorithm 1, the initial code and the decimal value increase by +1 until the algorithm reaches the desired value, forming the corresponding binary code.
**Algorithm 1**  Increment Code**Require:**bincodestr**Ensure:**bincodestr+1             ▹ The input incremented
1:p← position of the first bit 1 that has 0 in left2:aux←bincodestr[p]3:bincodestr[p]←bincodestr[p−1]4:bincodestr[p−1]←aux5:**if** bincodestr[p−2]=1 **then**         ▹ Verify out-of-bounds6:    left1← position of first bit 1 in the left of p−27:    Remove all bits 0 between p−2 and left18:    Prepend all 0 removed to bincodestr9:**end if**

The encoding region of each CM contains 78 bits, of which 8 are intended for checksum, leaving 70 bits for coding based on the ink level, which can have positive or negative values, as shown in Figure 3. Therefore, the marker can encode $2^{70}$ distinct forms of codes representing more than 1 million codes.

#### 3.2.1. Tape Generator

The web application *tape generator* (https://tape-generator.glitch.me/, accessed on 15 May 2024). facilitates the generation of the proposed marker for printing by simply drawing its positions on the floor plan of the location, as shown in Figure 4. When the user enters the dimensions, the system converts meters to pixels to calculate the *scale factor* for projecting the lines that represent the tape on the drawing. Each tape segment has an average of ≅0.23 m, distributed according to the length of the line drawn in the floor plan image.

The tape-generation process begins with loading a floor plan and information about the usable area of the building to apply the tape to implement the indoor location system. The next step is to draw lines on the floor plan image that correspond to the positioning of the tape on the property. Depending on the line size positioned on the floor plan, TMs with six, five, four, and three CMs will be generated. The rest of the line will be completed with individual CMs to cover the perimeter delimited by the user, as illustrated in Figure 5.

The tape-generation order follows the line drawn by the user in the web module, generating the TMs and CMs in ascending order. Hence, the first line drawn contains the first CMs, as illustrated in Figure 6.

At the end of the previous step, the user can download the tape in bitmap format to be printed. Furthermore, a configuration file in JSON format with metadata containing the IDs and coordinates of the CMs and TMs generated in the previous step can be saved and incorporated into the mobile module.

#### 3.2.2. Multiscale Functionality

The tape-shaped marker is designed to be robust against varying distances between the camera and the target. Thus, the user can move freely in space without losing the marker reading.

The system starts the TM detection process, displaying the IDs. If TM detection fails, the system starts detecting CMs, showing the IDs of the detected ones. The TM reading depends on the number of black pixels displayed on the tape. For reading to be possible, the user must be at a certain distance that frames an entire segment in the image with the four parallel bars on both sides that indicate the beginning and end of the TM segment. The following steps describe a simple algorithm for finding and reading the TM:Detect the horizontal lines in the tape, extracting only the most representative line that makes up the tape.Rotate the image to be parallel to the abscissa axis through angular adjustments according to the line detected in the previous step, leaving the detected line horizontal and cutting the image longitudinally.Apply grayscale, contrast, smoothing, and threshold filters.For the horizontal line of each image, the presence of patterns that indicate the beginning and end of the TM is verified. When detecting two patterns, convert the line into a bit string.Simplification of the bit string into unit values.Store the simplified bits in an array for voting.Apply voting to detected bit sequences by selecting the sequence with the highest frequency. If the confidence is greater than or equal to 70%, the string is returned. Otherwise, the algorithm returns null.

Line detection [65] starts with transforming the color image into grayscale, as shown in Figure 7a, and edge detection by applying a Canny filter with a minimum value of 230 and a maximum value of 255, as shown in Figure 7b. Then, the Hough Transform method [66] detects lines with parameters rho = 1, theta=π/180, and threshold>240, as shown in Figure 7c, which allows for the rotation and cropping of its surrounding area. This area may contain a TM. Finally, the algorithm extracts the bits with horizontal scanning from the binary image, as shown in Figure 7d.

Successful horizontal scans of the binary image participate in voting to ensure decoding reliability. For example, if the result found “10101011” seven times and three more different results, the decoder provides the result “10101011” with a reliability of 0.7, using a confidence of C≥0.5. When the reliability is below 0.5, the algorithm returns null for this iteration.

CM reading occurs when TM detection fails due to the lack of framing of the vertical bars, delimiting the beginning and end of the TM. The process begins with detecting the Finder Patterns, as shown in Figure 8a, then transforming the perspective of the coding region [65], as shown in Figure 8b, and extracting the bits from each CM on the tape, as shown in Figure 8c. The following steps describe the process:Applying preprocessing filters: grayscale conversion, contrast enhancement, and threshold filters.Extraction of the Finder Patterns square contours that delimit the region of each marker present on the tape, also extracting points corresponding to the region of the markers. Otherwise, the process flow returns to the beginning.Correction of the image perspective to represent the actual aspect ratio of the marker.Reading the grid pixels of the marker coding region in order to divide the image into cells. Each cell in the grid will correspond to the value one for pixels greater than 127, and for smaller values, the value 0 will be assigned in the bit matrix.Validation of the pixels that represent the checksum bits.The resulting bit output will be converted into a bit vector, selecting only the bits in the coding region. The process repeats until the algorithm extracts the last pair of Finder Patterns.Returns the vector of detected bits.

The reading of CMs is successful when the perspective correction allows the Finder Patterns to fit correctly in the reading grid. This correction allows for a consistent bits extraction in the encoding region and the extraction of the checksum bits that contain the encoding region information hashed by the BLAKE2 function [67]. When reading the code, the checksum must coincide with the hashed value read in the encoding region, validating the value of the encoding region. Otherwise, the CM read fails, and the algorithm goes to the next detected CM.

#### 3.2.3. Mobile Application

The mobile application was developed for Android version 5 or higher using the Kivy framework, a free and open-source Python framework for developing mobile applications and other multitouch application software with a natural user interface [68]. In addition to the Kivy framework, the application uses the OpenCV computer vision and image-processing library [69].

The application allows for the reading of the tape with the detection of TMs and CMs, storing the detections in a JSON file to display the location, in addition to capturing and exchanging the rear camera for the front, as well as changing the camera orientation to portrait or landscape.

## 4. Experiments and Results

This section aims to present the experiments with the tape-shaped marker to analyze its strengths and weaknesses. The tests were carried out in a 3D simulation environment and in a real environment with good lighting conditions.

The experiments in a 3D simulation environment made it possible to create a controlled environment to analyze the accuracy and improvements of the tape-shaped marker. After that, we will dedicate efforts to the analysis of a mobile application for an indoor location in conditions of a real environment with variations in lightness, blurring, and distance. The source code and datasets employed in the experiments are available for other researchers to reproduce these experiments in (https://github.com/BeneditoSRNeto/Tape-shaped-markers, accessed on 15 May 2024).

### 4.1. Simulation in a 3D Environment

This subsection presents the experimental tests in a 3D simulation environment to analyze the performance of the tape-shaped marker, starting with the accuracy test and followed by the performance in the localization test.

Initially, we performed the experiments in a 3D simulation environment developed in Blender 4.0, as shown in Figure 9. Detection tests consider the STag [18], QRCode [64], and ArUco [17] markers at different distances to analyze the accuracy of the proposed marker.

STag [18] features a black background and a white circle and is robust against difficult viewing angles and varying lighting. ArUco [17] features a black background and white bars, and its detection algorithm is robust against partial occlusion and lighting variations. QRCode [64] has a white background and black bars and can be encoded in its body with a large volume of bytes, in addition to being sensitive to lighting and occlusion. The marker-detection algorithms ran in the Python programming language with the OpenCV [69] library on a laptop computer with a 1.80 GHz Intel®Core™i7-8565U CPU with Windows 11 operating system.

A tape with six CMs was generated for the accuracy test, measuring 6 cm in height. In the same way, six QRCodes, six STags, and six ArUcos were generated with 6 × 6 cm dimensions. The markers were positioned on the wall of the 3D environment to capture images ranging from 10 m to 0.5 m. For each distance, six images were captured. Accuracy was calculated by S/(S+F), and Success *S* occurs when the marker is detected correctly. The *F* failure occurs when the algorithm fails to identify the IDs or find the marker.

Figure 10 shows the tests that evaluated the detection accuracy of the proposed marker with STag [18], QRCode [64], and ArUco [17], demonstrating that the tape-shaped marker is detected at all distances ranging from 10 to 0.5 m, demonstrating its robustness at multiple distances. Figure 11 shows the reading accuracy of the proposed marker with details of the TM (blue line) and CM (red line). The CM reading performs better than the TM when the distance is short. The accuracy is almost the same between 2 and 6 m, varying slightly. When the distance is more than 6 m, the TM reading performs better than the CM.

The STag [18] readings, shown in the purple line (Figure 11), fail at a distance of 3 m. The QRCode [64] readings, shown in the orange line (Figure 11), fail at a distance of 3.5 m. The ArUco [17] readings, shown in the green line (Figure 11), fail at a distance of 5.5 m. The CM’s readings (Figure 11) fail at a distance of 7.0 m. TMs cannot be detected at a distance of 0.5 to 1.5 m because the camera cannot completely frame the size of the TM. Then, the detection’s CMs and TMs complement each other for these distances. Even with variations in the TM’s accuracy, it is effective for distances greater than 1.5 m, where it was possible to frame the camera and continue the detection service.

After testing the accuracy of the markers at different distances, a test was carried out that assesses the robustness against light variation. The experiment was based on the power in watts of the luminosity efficiency of a point of white light. The light point was positioned in the 3D environment at a distance of 2 m from the markers. The camera was positioned at the same distance from the lighting point to capture the images. The tape-shaped, STag [18], QRCode, and [64] ArUco [17] markers were exposed to lighting that varied from 0 to 50,000 watts, as shown in Figure 12.

Figure 13 shows the result of the experiment with the tape-shaped, STag [18], QRCode, and [64] ArUco [17] markers. The experiment with the tape-shaped marker shows that it was not possible to detect the tape-shaped marker with the lighting of 0 watts of light power. Only by exporting lighting from 4 w to 1600 w was it possible to detect the tape-shaped marker. STag [18] can be detected at lighting exposure from 0w up to 40,000 w. QRCode [64] and ArUco [17] were detected at lighting exposure from 0 w to 30,000 w.

After the robustness test against lighting variations, a test was carried out to evaluate the robustness against viewing angles with the markers STag [18], QRCode [64], ArUco [17], and with the tape-shaped marker. Similar to the accuracy test at different distances, a tape with six CMs was generated, measuring 6 cm in height. In the same way, six QRCodes, six STags, and six ArUcos were generated with 6 × 6 cm dimensions. The camera was positioned at angles ranging from 0° to 90° from the target on the X axis, at a distance of 3 m, as shown in Figure 14.

Figure 15 shows that the tape-shaped marker managed to maintain the reading up to the 40° angle. ArUco and STag were robust up to the 70° angle. QRCode was detected up to an angle of 30°.

After robustness tests at difficult viewing angles, an indoor localization test was carried out with the STag [18], QRCode [64], and ArUco [17] markers and with the tape-shaped marker. The markers were positioned in a 3D environment with two rooms. Room “A” with the following measurements: 3.7 m wide and 4.0 m long with a 300 w lighting point. Room “B” that is 3.7 m wide and 2.2 m long and with a 150 w lighting point. The test environment has a path that the camera followed between room “A” and room “B” with variation in light and shading, as shown in Figure 9. The route generated videos for each test with 100 frames with a 1080 × 1920 pixels resolution, which simulate the video resolution of a smartphone in portrait camera orientation.

The tape-shaped marker was generated in the tape-generator web application, as shown in Figure 4. When drawing lines on the floor plan of the test environment, a total of 76 CMs and 13 TMs were generated, 6 cm high and with a width according to the dimensions of the perimeter of the environment. Furthermore, 76 QRCodes [64], 76 STags [18], and 76 ArUcos [17] were generated with 6 × 6 cm dimensions.

To test again the robustness against difficult viewing angles now in the localization test, the tape-shaped markers, STag [18], QRCode [64], and ArUco [17] were positioned in the test environment under the following positioning conditions: (A) markers positioned on the wall baseboard; (B) markers positioned at camera height; and (C) markers positioned on the floor. Figure 16 illustrates the positioning of the markers in the indoor localization test.

The tests in condition “A” were carried out with a perspective camera with a focal length of 50 mm, positioned 140 cm from the floor, and an angle of −30° on the X axis. For the test under condition “B”, it was again the perspective camera with a focal length of 50 mm, positioned 140 cm from the floor, and with an angle of 0° on the X axis. For the test under condition “C”, the perspective camera with a focal length of 50 mm was used again, positioned 140 cm from the floor and with an angle of −30° on the X axis.

Figure 17, Figure 18 and Figure 19 illustrate the detection points during the journey from room A to room B with the markers positioned in the environment under conditions “A”, “B”, and “C”. The triangles indicate the camera direction and movement, and their size indicates the time taken to detect the marker; the larger the triangle, the longer the time to detect the marker. The absence of triangles on the route indicates detection failure.

Figure 17a shows that the tape-shaped marker under positioning condition “A” (gray triangles) had 16 detection points, with a detection time of less than 24171 ms. The performance of ArUco [17], in Figure 17b, shows (green triangles) that it had 19 detection points with a detection time of less than 1343 ms. The performance of QRCode [64], in Figure 17c, shows (orange triangles) that it had one detection point with a detection time of 2562 ms. The performance of STag [18], in Figure 17d, shows (purple triangles) that there were 12 detection points with a detection time of less than 1468 ms.

Figure 18a shows that the tape-shaped marker in the positioning condition “B” (gray triangles) had 18 detection points with a detection time of less than 9421 ms. The performance of ArUco [17] (Figure 18b) shows (green triangles) that it had 18 detection points with a detection time of less than 1187 ms. The performance of QRCode [64] (Figure 18c) shows (orange triangles) that it had 11 detection points with a detection time of less than 2218 ms. The performance of STag [18] (Figure 18d) shows (purple triangles) that there were 17 detection points with a detection time of less than 1218 ms.

Figure 19a shows that the tape-shaped marker in the positioning condition “C” (gray triangles) had 12 detection points, as it was not possible to read the CM’s child markers, only the TMs with a detection time of less than 54859 ms. The performance of ArUco [17] (Figure 19b) shows (green triangles) that it had 19 detection points with a detection time of less than 1375 ms. QRCode’s performance [64] (Figure 19c) was not detected under positioning condition “C”. The performance of STag [18] (Figure 19d) shows (purple triangles) that there were 14 detection points with a detection time of less than 1406 ms.

The visualization of the data regarding the time of marker detections in the localization test is shown in Figure 20 under the marker positioning conditions. In condition A, the median time for detections of the tape-shaped marker was 375 ms. The median time for ArUco marker detections [17] was 15.63 ms. The median detection time for the QRCode marker [64] was 2562.5 ms. Finally, the median time for STag marker detections [18] was 46.88 ms.

In condition B, the median time for detections of the tape-shaped marker was 406.25 ms. The median detection time for the ArUco [17] marker was 46.88 ms. The median detection time for the QRCode marker [64] was 46.88 ms. Finally, the median time for STag marker detections [18] was 31.25 ms.

In condition C, the median time for detections of the tape-shaped marker was 3484.38 ms. The median detection time for the ArUco [17] marker was 46.88 ms. The median time for detections of the QRCode marker [64] was not calculated, as there were no detections. Finally, the median time for STag marker detections [18] was 31.25 ms.

Table 3 shows the minimum, maximum, average, and median values of the computation of the time of detection of the tape-shaped marker, STag [18], QRCode [64], and ArUco [17] carried out during the indoor localization test in the 3D simulation environment under test conditions A, B, and C.

### 4.2. Mobile Device Testing

This subsection presents experimental tests in a real environment using an application on a smartphone and printed tape-shaped markers positioned in the environment to analyze the performance of the location test.

After testing in a simulated environment, tests were carried out in a real environment with an application embedded in the Xiaomi Redmi Note 9 s smartphone, which has the Android 11 operating system. It has a 48-megapixel main camera with a resolution of 8000 × 6000 pixels and a camera aperture of F 1.79 + F 2.2 + F 2.4 + F 2.4, digital stabilization, and autofocus.

The tape segments were printed on A4 paper and pasted in ascending order as designed in the “Tape generator” web application (Figure 4). Then, they were positioned on the footer of the test environment for the reading process according to the user’s path through the environment, as shown in Figure 21.

The user traveled the route using the smartphone without sudden movements, as shown in Figure 22a. He used the smartphone with a tripod on a chair to keep the camera as stable as possible, as shown in Figure 22b, and used the smartphone with a Gimbal stabilizer, as shown in Figure 22c. In both cases, the base of the smartphone was located at a distance of 140 cm from the ground, with the camera in portrait mode. The route was segmented into 20 stopping points, each lasting 10 s and taking 5 s to travel between points.

Figure 23 shows that the median time to detect markers with the smartphone in the user’s hand was ≅305 ms. With the smartphone on the chair, it took ≅290 ms. With the smartphone on the Gimbal, it took ≅260 ms.

## 5. Discussion

The experimental tests measured the performance of the tape-shaped marker with readings at multiple distances compared to ArUco [17], QRCode [64], and STag [18]. This shows that the tape-shaped-marker-detection algorithm is robust at distances of 10 to 0.5 m, similar to the Fractal Marker [46], HArCo marker [25], and AprilTag 3 [30]. However, the tape-shaped marker has several child markers distributed horizontally, allowing them to be read from a single distance, unlike the Fractal Marker [46], HArCo marker [25], and AprilTag 3 [30], which have concentric child markers read according to the distance from the camera.

The tape-shaped marker allows you to encode $2^{70}$ CM distinct codes that can be used in large areas and can also be mapped to each floor of a building or shopping center; in addition, the marker can be flexible in the number of child marker CMs, as it is not limited to just 6 CMs but can have more CMs present in a TM, increasing the number of different TM codes. For example, adding 10 CMs in a TM would be $2^{10}$ combinations of different TMs, where each CM is 0.23 m, and by multiplying by 10, each TM can cover 2.3 m, and multiplying 2.3 by $2^{10}$ covers a perimeter of approximately 2355 meters.

In terms of the coverage area of the tape-shaped marker, comparing the use of fiducial markers with Wi-Fi technology in a location system, the Wi-Fi signal can be used to cover large areas; however, it may have attenuation or a loss of signal due to signal obstruction or interference from other Wi-Fi signals, causing location inaccuracy.

The localization system with fiducial markers does not generate location ambiguities as can occur with Wi-Fi technology; for example, it may not be able to distinguish whether a person is close to a wall on the inside or outside the room. Furthermore, the Wi-Fi device needs to be connected to electricity to function. If there is a lack of electricity, the Wi-Fi device becomes inoperative and the location system stops working. Fiducial markers do not depend on electricity to perform their function in a location system. However, it can suffer physical degradation due to humidity and exposure to sunlight.

The location tests with the tape-shaped marker in the 3D environment allowed us to analyze the detection time during the journey from one room to another under positioning conditions “A”, “B”, and “C” (Figure 16). The results demonstrated that in conditions “A” and “B”, the marker performed well in detections. There were few detections in condition “C” due to the challenging viewing angle. This result demonstrated the need for improvements in the process and extraction of features at difficult viewing angles, emphasizing detecting CMs that were not detected in the condition “C” positioning.

When measuring the time that the algorithm took to detect the tape-shaped marker, it was considered high compared to the ArUco [17], QRCode [64], and STag [18] markers. This required optimization in the algorithm of detection to be able to be detected with an average time of close to ≅200 ms.

Tests in the real environment with the smartphone showed that the average detection time (Figure 23) was shorter than tests in the 3D environment (Figure 20) due to the application already being in binary code executed on an Android device. Furthermore, the stability of the camera’s movement directly influences the marker-detection process, as when the smartphone is in the user’s hands, the average time to perform a detection can last ≅305 ms. When the smartphone is on a tripod on a chair, the average time to perform a detection can reach up to ≅290 ms, and when the smartphone is on a Gimbal, the average time to perform a detection can reach up to ≅260 ms.

Considering that the route of the location test with the smartphone had 20 stopping points, each stop was limited to 10 s. At some stopping points, it was not possible to carry out detection due to the blurring of the camera on the route, demonstrating that the algorithm for tape-shaped marker detection is not robust against blurring. However, if the dwell time at each point was longer than 10 s, the camera would become stable again with another frame readout, and tape playback could be resumed.

As for robustness against blurring, the algorithm based on geometric calculations will be implemented, as in the works of RuneTag [53], BlurTags [32], monospectrum markers [22], PRASAD et al. [52], and WhyCode [60]. They are markers that present neutral regions, interspersed elements, or spacing between internal elements. They tend to be easier to detect even with blurring as they allow the internal elements of the marker to have high contrast, allowing for detection even with a blurred image.

For robustness against difficult viewing angles, we used an algorithm found in markers: Standard Pattern [55], TRIP [59], SIFT and SURF [28], AprilTag [16], AprilTag 2 [29], RuneTag [53], BlurTags [32], Pi-Tag [51], color marker-based [39], ArUco [17], BullsEye [33], PRASAD et al. [52], WhyCode [60], HArCo marker [25], STag [18], Jumarker [21], SVMS [56], X-tag [61], AprilTags 3D [31], Farkas et al. [45], Cantag [34], and Order Type Tags [50]. They use the homographic matrix to correct the image perspective of the marker by detecting corners or ellipses to enable detection at difficult viewing angles.

For robustness against partial occlusion, the detection of segments of the tape (child elements) allows for this type of robustness. Even if a part of the tape is occluded, other CMs present on the tape will be available for reading, allowing the tape to continue to be detected. This type of robustness was implemented based on detection algorithms that were robust against partial occlusion found in markers: ReacTIVision [23], AprilTag [16], AprilTag 2 [29], RuneTag [53], CoP-Tag [42], Pi-Tag [51], ArUco [17], CCTag [35], HArCo marker [25], Topotag [58], LFTag [48], Jumarker [21], ICL [47], SVMS [56], X-tag [61], Order Type Tags [50], sSLAM [20], and the Fractal Marker [46].

For robustness against lighting variations, the detection algorithm made use of the adaptive threshold, allowing the tape to be read even with low ambient lighting, similar to the detection algorithms used in ArUco [17] markers, TRIP [59], ReacTIVision [23], FourierTag [27], AprilTag [16], AprilTag 2 [29], Seedmarkers [54], AprilTags 3D [31], Chilitags [36], Cantag [34], RUNE-Tag [53], Pi-Tag [51], BullsEye [33], WhyCode [60], STag [18], Topotag [58], SVMS [56], and ICL [47].

It was possible to verify, in both test environments, the efficiency of reading on multiple scales, in a controlled and uncontrolled manner, of the variations in environmental conditions in order to verify that the tape detection occurs with a minimum illumination of 4 W and a maximum of 1600 W of luminous power; that the limit of the detection angle was up to 40°; and that it was possible to read the tape-shaped marker from 10 to 0.5 m away. These tests confirm the main characteristics of this fiducial marker system, which minimizes discontinuity in the location service. As for the use of different mobile devices to detect the marker, it only differentiates the results in the stability of the camera, especially when using lower processing and camera configurations than those used in the test, as image blurring may occur when the camera is moved around the environment so that the marker cannot be read. Except for this condition, the other results will be generic and replicable for other mobile devices.

In addition to this fiducial marker system being able to be embedded in mobile devices, it can be combined with other sensors such as Wi-Fi, Bluetooth, accelerometer, gyroscope, and other sensors present in smartphones, robots, or drones to improve the efficiency of localization systems.

## 6. Conclusions

Using fiducial markers in an internal location system allows for excellent cost benefits in implementation due to the ease of using a smartphone and visual marks with coded information positioned in a scenario. Through the tape-shaped marker, it was possible to have a marker with a linear appearance positioned around the entire perimeter of the environment. The information coded sequentially allows for discontinuity to be minimized in the location service.

With the multiscale detection algorithm, the tape-shaped marker can be expanded to larger environments such as squares, shopping centers, subway stations, airports, and museums. It has a range of up to 10 m for a 6 cm tall tape with an aspect ratio of 25:7. Increasing the height will increase the marker’s reading distance to suit these types of environments.

Another advantage of using a marker as a tape is the ease of configuring it remotely to the environment. The tapes can be generated in a web application with an image of the location’s floor plan, avoiding being present on-site to take measurements of the perimeter. This is unlike the configuration setup of a system with real markers, where there is a need to be present to record images in various aspects of the environment. Elements such as tables, chairs, doors, and windows in the captured scene are subject to change.

The disadvantage of implementing the tape-shaped marker was printing the segments and sticking them in an orderly and aligned manner throughout the environment; in addition, the marker can be degraded by humidity and exposure to sunlight.

The proposed marker proved robust against variations in distance from the target, difficult viewing angles, ambient lighting conditions, and partial occlusions. However, the tape-shaped-marker-detection algorithm showed a low performance, with a high reading time compared to markers available in the literature such as ArUco [17], QRCode [64], and STag [18]. Hence, the implementation of the locator and recognizer is in the prototypical stage and needs improvements in its performance, as this work needs many financial and human resources for its development. This work focuses on a new approach to fiducial marker morphology but can be improved in terms of code and time performance in further works.

In future work, the blur-robust detection algorithm will be implemented to allow reading in conditions of camera movement and improvements in robustness against difficult viewing angles. Since CM markers could not be read in the condition of C positioning, only the TMs were read, even when applying image-perspective-correction techniques. Therefore, there is a need for studies to identify at what angle the marker can be detected. In addition, a study will be carried out to apply the customization of morphological styles to the tape-shaped marker to give it an environmentally friendly appearance without losing its main characteristics, such as the construction work presented by Zhang et al. [12].

Therefore, this study showed a fiducial marker based on computer vision techniques based on the geometric calculations algorithm. This allows for detection at multiple distances with a low implementation cost, keeping the location service continuously operational because the marker can be present everywhere along the perimeter of the environment.

## Figures and Tables

**Figure 2 sensors-24-04605-f002:**
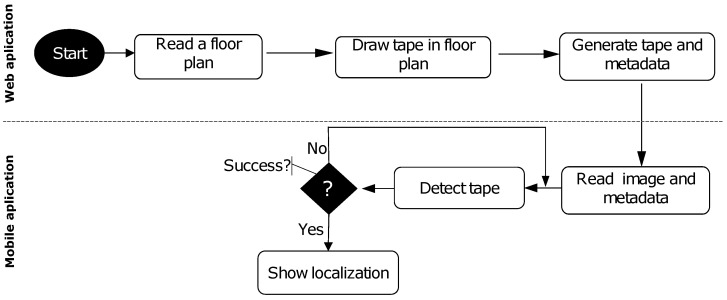
Activity flow of the system of the tape-shaped markers with a web module and mobile module.

**Figure 3 sensors-24-04605-f003:**
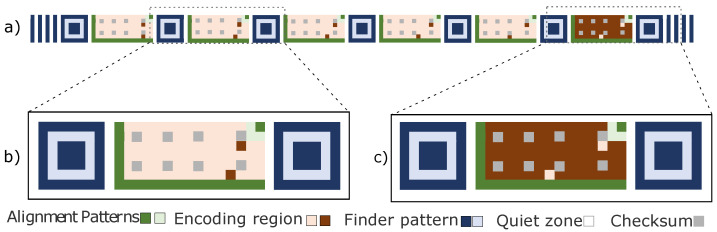
The marker’s design. (**a**) Encoding for TM larger-scale detection. (**b**) Encoding for CM positive values. (**c**) Encoding for CM negative values.

**Figure 4 sensors-24-04605-f004:**
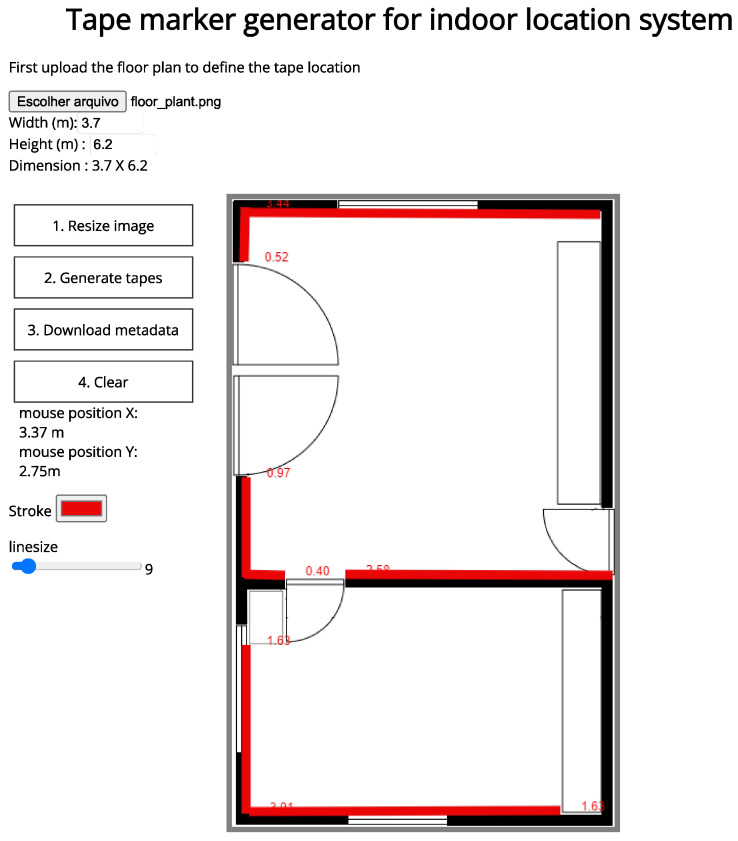
Bitmap of the floor plan with 2 rooms in the tape generator application. In red, the projection of the tape on the perimeter of the room.

**Figure 5 sensors-24-04605-f005:**
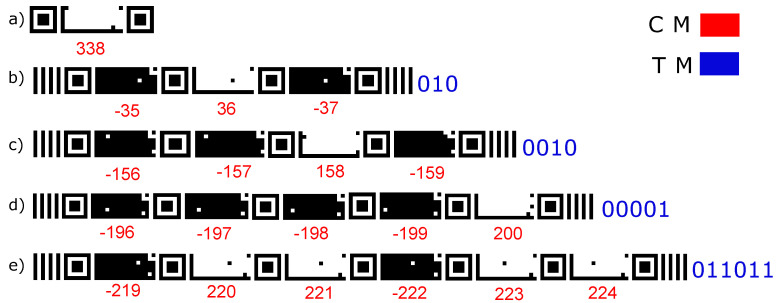
Examples of tape shapes with (**a**) 1 CM, (**b**) 3 CMs, (**c**) 4 CMs, (**d**) 5 CMs, and (**e**) 6 CMs.

**Figure 6 sensors-24-04605-f006:**
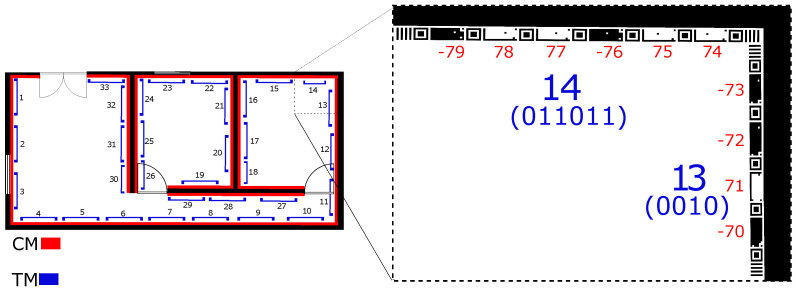
This is an example of a floor plan containing the tape (in red) for each room with its respective CM codes. The blue square brackets show the TMs formed by a set of 6 CMs.

**Figure 7 sensors-24-04605-f007:**
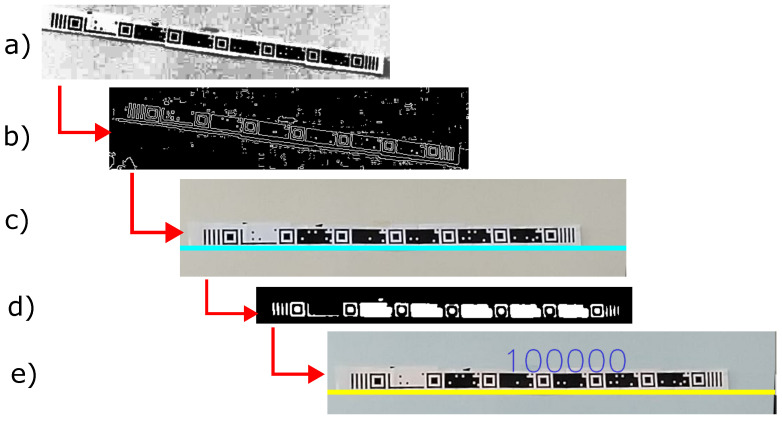
TM detection process. (**a**) Grayscale image. (**b**) Canny filter. (**c**) Rotated 180 degrees. (**d**) Binarized image. (**e**) Detected TM image.

**Figure 8 sensors-24-04605-f008:**
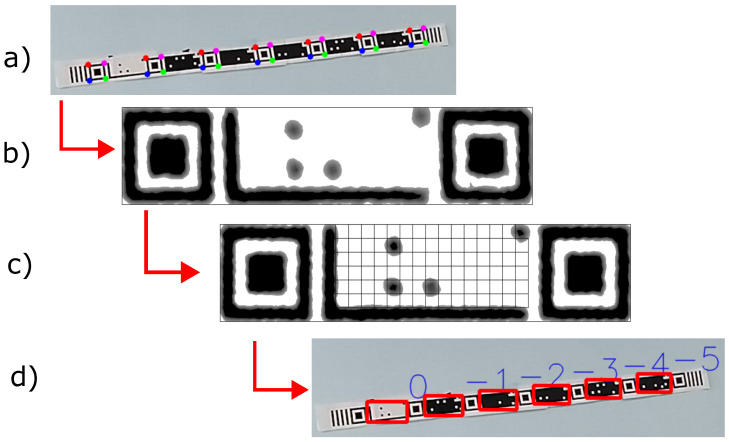
CM detection process. (**a**) Detection of Finder Patterns, (**b**) image warp, (**c**) grid on encoding region, (**d**) detected CM’s image.

**Figure 9 sensors-24-04605-f009:**
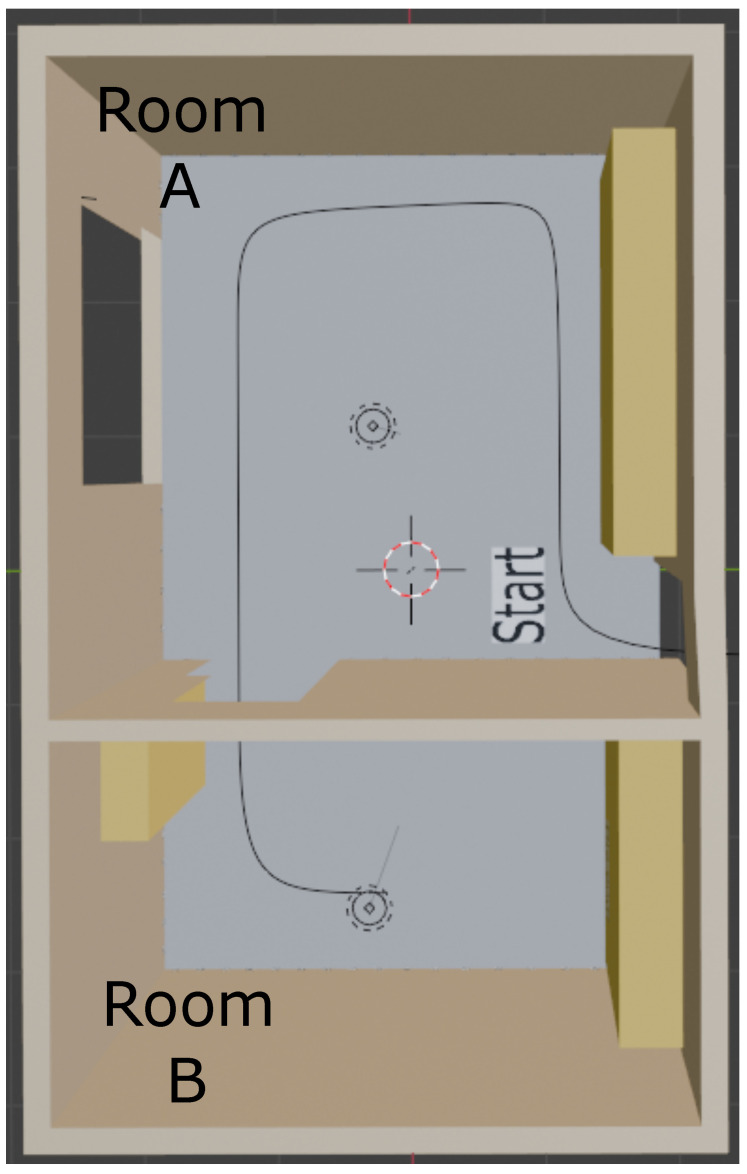
Visualization of the test environment simulation with the route from room A to room B built in Blender 4.0.

**Figure 10 sensors-24-04605-f010:**
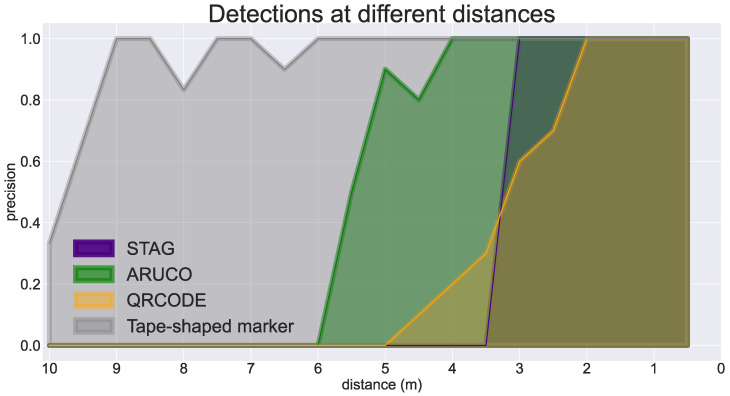
Reading accuracy at distances ranging from 10 to 0.5 m, in purple with the STag marker, in green with the ArUco marker, in yellow with the QRCode marker, and in gray the performance with the tape-shaped marker.

**Figure 11 sensors-24-04605-f011:**
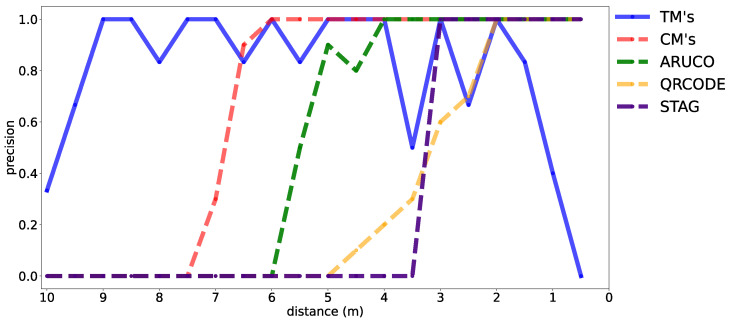
Detail of the performance of the tape-shaped marker when reading at distances ranging from 10 to 0.5 m, in blue for reading the TM, in red for the CM, in green for ArUco, in yellow for QRCode, and in purple for STag.

**Figure 12 sensors-24-04605-f012:**
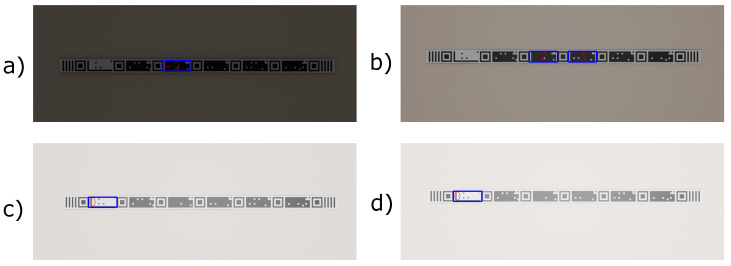
Some images of the tape-shaped marker used in the robustness against lighting variation experiment. (**a**) Exposed to 4 w lighting. (**b**) Exposed to 100 w lighting. (**c**) Exposed to 1000 w lighting. (**d**) Exposed to 1500 w lighting.

**Figure 13 sensors-24-04605-f013:**
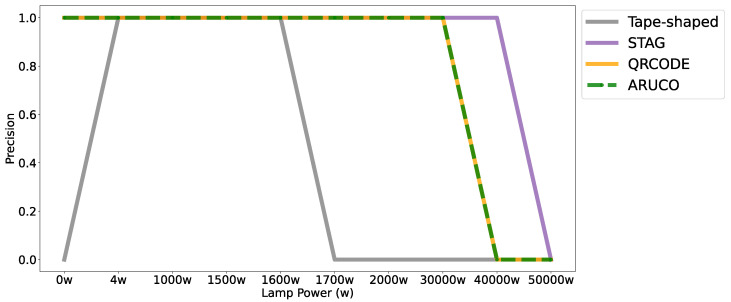
Accuracy of markers with lighting variation from 0w to 50,000 w; in gray is the performance with the tape-shaped marker, in purple the STag marker, in yellow the QRCode marker, and in green the ArUco marker.

**Figure 14 sensors-24-04605-f014:**
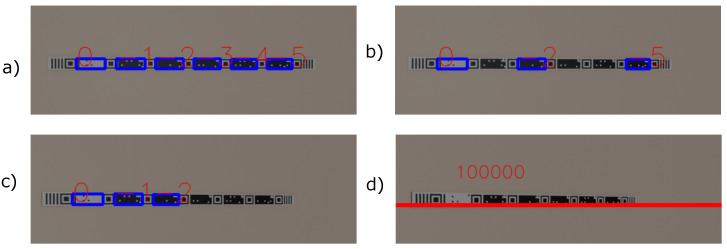
Markers at difficult viewing angles. (**a**) Tape-shaped marker with an angle of 10°. (**b**) Tape-shaped marker with an angle of 20°. (**c**) Tape-shaped marker with an angle of 30°. (**d**) Tape-shaped marker with an angle of 40°.

**Figure 15 sensors-24-04605-f015:**
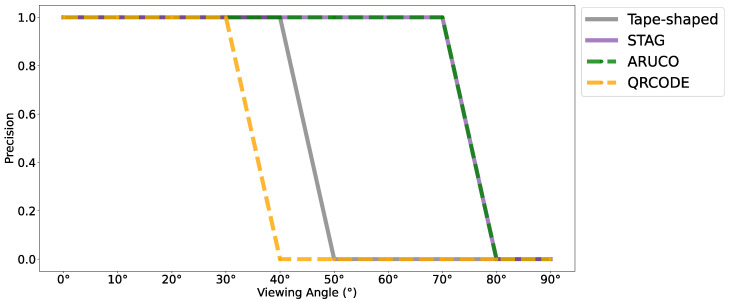
Accuracy of markers at difficult viewing angles from 0° to 90°; in gray is the performance with the tape-shaped marker, in purple the STag marker, in green the ArUco marker, and in yellow the QRCode marker.

**Figure 16 sensors-24-04605-f016:**
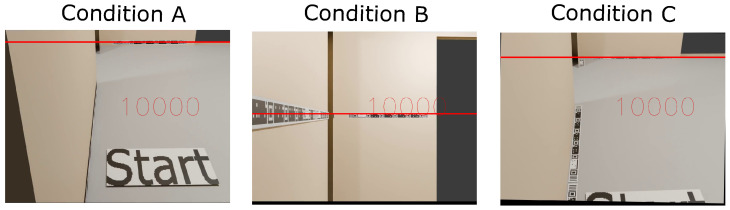
Positioning of the markers in the test environment in the conditions, with the marker at the base of the wall (condition A), with the marker on the wall at a height of 140 cm at the floor (condition B), and with the marker on the floor close to the wall (condition C).

**Figure 17 sensors-24-04605-f017:**
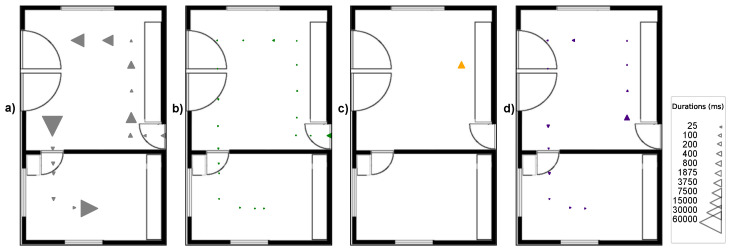
Detection points on the route from room A to room B in condition “A” of marker positioning. (**a**) Tape-shaped marker. (**b**) ArUco. (**c**) QRCode. (**d**) STag.

**Figure 18 sensors-24-04605-f018:**
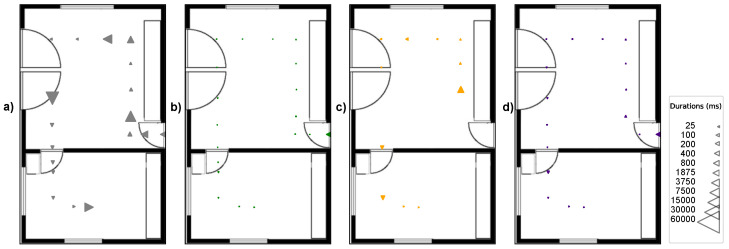
Detection points on the route from room A to room B in condition “B” of marker positioning. (**a**) Tape-shaped marker. (**b**) ArUco. (**c**) QRCode. (**d**) STag.

**Figure 19 sensors-24-04605-f019:**
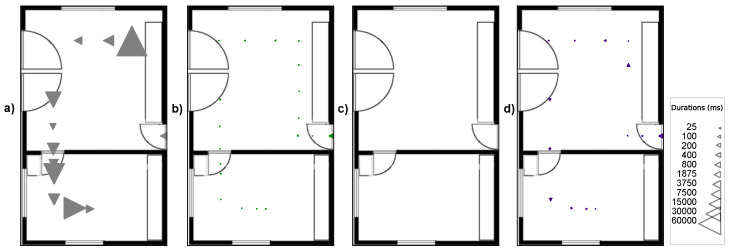
Detection points on the route from room A to room B in condition “C” of marker positioning. (**a**) Tape-shaped marker. (**b**) ArUco. (**c**) QRCode. (**d**) STag.

**Figure 20 sensors-24-04605-f020:**
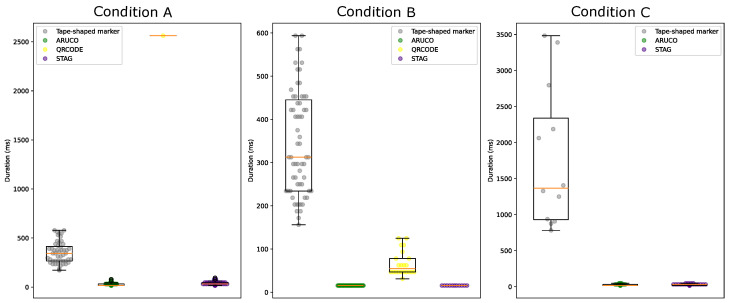
Duration time of detections on the route from room A to room B under conditions “A”, “B”, and “C” using the ArUco, QRCode, STag, and tape-shaped markers.

**Figure 21 sensors-24-04605-f021:**
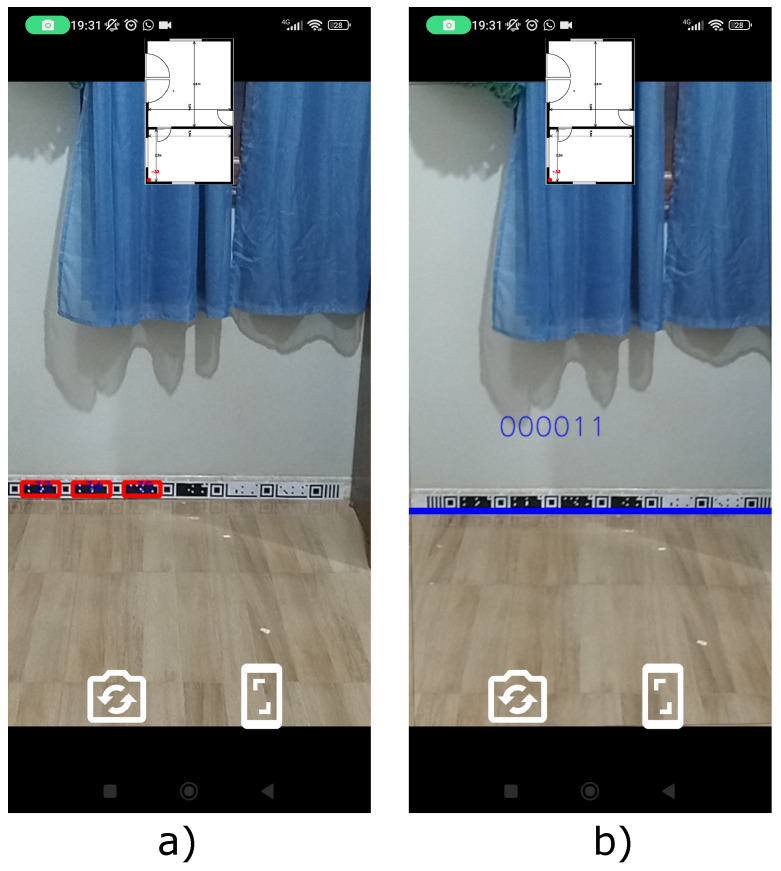
Running the location test application on the smartphone. (**a**) Detection of CMs. (**b**) Detection of TMs.

**Figure 22 sensors-24-04605-f022:**
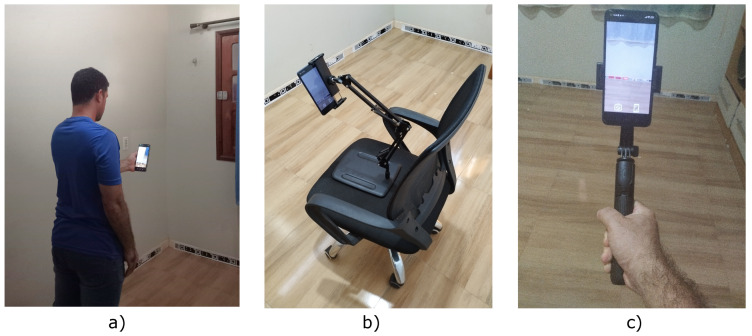
Camera conditions are tested for localization. (**a**) Smartphone in the user’s hand. (**b**) Smartphone on the tripod. (**c**) Smartphone on the Gimbal.

**Figure 23 sensors-24-04605-f023:**
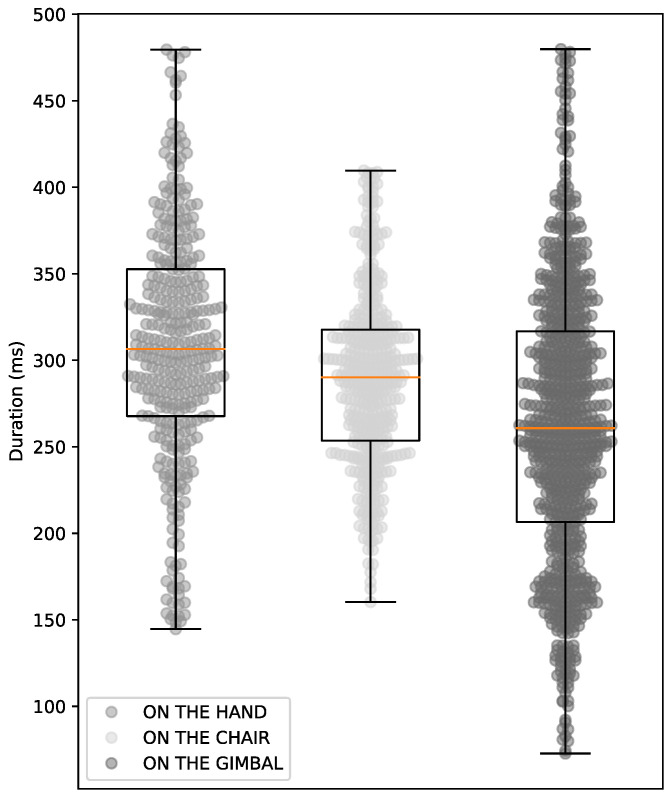
Duration of detections on the condition of smartphone.

**Table 2 sensors-24-04605-t002:** Illustration of CM numeric encoding with 4 bits.

Decimal	WBC	Proposed Code	Ink Proposed	Level ink
0	0000	0000		less
1	0001	0001	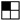	less
2	0010	0010	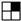	less
3	0011	0100	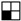	less
4	0100	1000	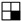	less
5	0101	0011	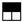	cannot be used
6	0110	0101	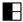	cannot be used
7	0111	1001	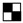	cannot be used
8	1000	0110	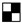	cannot be used
9	1001	1010	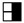	cannot be used
10	1010	1100	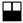	cannot be used
11	1011	0111	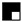	more
12	1100	1011	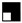	more
13	1101	1101	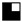	more
14	1110	1110	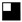	more
15	1111	1111	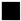	more

**Table 3 sensors-24-04605-t003:** Computation of the time of detections that occurred in the indoor location test under test conditions A, B, and C.

	Condition A		Condition B		Condition C	
	AVG	MED	AVG	MED	AVG	MED
Tape-shaped	1458.07	375.00	785.00	406.25	8242.53	3484.38
ArUco	43.88	15.63	91.57	46.88	91.57	46.88
QRCode	2562.5	2562.5	91.57	46.88	-	-
STag	91.57	46.88	67.97	31.25	110.58	31.25

## Data Availability

Data are contained within the article.

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
