# Peer review of "Tape-Shaped, Multiscale, and Continuous-Readable Fiducial Marker for Indoor Navigation and Localization Systems"

_sensors, 2024, doi:10.3390/s24144605_

Round 1

Reviewer 1 Report

Comments and Suggestions for Authors

The research paper "Indoor localization system based on fiducial Tape-shaped markers" is presented as an approach to indoor location systems using computer vision. The approach focuses on using tape-shaped markers for continuous operational performance.

Overall, the paper is well-written and easy to follow. The introduction and related work could be improved with a better context and discussion regarding the performance, benefits and challenges of the main indoor positioning approaches and technologies. 

The marker tape-based approach presented in this paper, in my opinion, has novelty and potential for some use cases. The technical details regarding the marker design, testing procedures, and experimental results are generally well presented, with subsections addressing different aspects of the approach. The experiments add value to the work, including both simulation and real-world tests. Publishing the code and datasets used in the experiments is commendable, promoting fairness and reproducibility.

My major concern is related to how this work is positioned and presented. From the title and abstract, I was expecting to find a complete localization system. However, after reading the paper and considering what is presented, I can only conclude that this solution is still far from being considered an "indoor localization system," as the title and abstract suggest. The paper lacks important details and experiments to be presented as such. For example, there is no information on how the position of the user is estimated or results regarding the localization/positioning performance (e.g., positioning error).

Essentially, the paper focuses on presenting a solution that can detect tape-shaped markers, and the results focus on detection accuracy and time. Therefore, in my opinion, this work should not be positioned as an "Indoor Localization System" unless these aspects are presented.

Aspects that are important to discuss, clarify or improve:

- The authors mention the benefits of their approach, however, they do not delve into the limitations and challenges, nor do they include a discussion or comparison with other indoor positioning technologies such as Wi-Fi or BLE in terms of expected positioning performance, deployment effort, or target use cases.

- What kind of localization/positioning accuracy is expected from this solution? 10cm? 1m? 10m?

- How does it compare in terms of deployment effort? This is a crucial aspect to clarify and consider, especially for large spaces such as shopping malls and airports, which are mentioned as potential application scenarios. Do the benefits outweigh the deployment effort when compared, for example, with the use of Wi-Fi that uses already existing infrastructure? Wi-Fi systems can cover larger areas with fewer access points, providing broader coverage with less physical deployment effort. Physically placing and aligning markers throughout the environment is time-consuming, especially in larger spaces. The coverage area of the markers may be limited, potentially requiring more markers for comprehensive coverage. For example, in large open areas (airports, shopping malls), the positioning performance will degrade or be impossible to estimate.

- What are the maintenance requirements or calibration procedures for the tape-shaped markers over time?

- The authors refer that the detection algorithm is robust against lighting variations, however, it does not address challenges related to other environmental factors such as shadows, reflections, or occlusions.

A better discussion of these aspects is in my opinion essential to clarify which use cases this approach would be viable or better than other approaches.

Comments on the Quality of English Language

Overall, the paper is well-written and easy to follow. 

Author Response

Review 1

The research paper "Indoor localization system based on fiducial Tape-shaped markers" is presented as an approach to indoor location systems using computer vision. The approach focuses on using tape-shaped markers for continuous operational performance. Overall, the paper is well-written and easy to follow. The introduction and related work could be improved with a better context and discussion regarding the performance, benefits and challenges of the main indoor positioning approaches and technologies. The marker tape-based approach presented in this paper, in my opinion, has novelty and potential for some use cases. The technical details regarding the marker design, testing procedures, and experimental results are generally well presented, with subsections addressing different aspects of the approach. The experiments add value to the work, including both simulation and real-world tests. Publishing the code and datasets used in the experiments is commendable, promoting fairness and reproducibility.

 R : OK, thank you!

My major concern is related to how this work is positioned and presented. From the title and abstract, I was expecting to find a complete localization system. However, after reading the paper and considering what is presented, I can only conclude that this solution is still far from being considered an "indoor localization system," as the title and abstract suggest. The paper lacks important details and experiments to be presented as such. For example, there is no information on how the position of the user is estimated or results regarding the localization/positioning performance (e.g., positioning error).

Essentially, the paper focuses on presenting a solution that can detect tape-shaped markers, and the results focus on detection accuracy and time. Therefore, in my opinion, this work should not be positioned as an "Indoor Localization System" unless these aspects are presented.

  R: We agree that our work does not present a localization system. We made a mistake when writing the title and abstract, as the aim of the work is to present a tape-shaped marker, a new fiducial marker design, not a localization system itself. In this sense, the title, abstract, and other parts of the text were rewritten.

Aspects that are important to discuss, clarify or improve:

- The authors mention the benefits of their approach, however, they do not delve into the limitations and challenges, nor do they include a discussion or comparison with other indoor positioning technologies such as Wi-Fi or BLE in terms of expected positioning performance, deployment effort, or target use cases.

R: A thorough comparison with other technologies would be beyond the scope of the work, as the purpose of this work is to propose and validate a new fiducial marker. Nevertheless, new tests were included with new environmental lighting conditions and viewing angles to evaluate in detail the performance of the marker proposed. These tests were added in lines 402 to 424 of the Experiments and Results section.

- What kind of localization/positioning accuracy is expected from this solution? 10cm? 1m? 10m?

 R: Considering that the algorithm implemented to recognize the proposed marker is a prototype and can be improved in accuracy in further works that focus on the algorithm and recognition performance, that is not strictly the aim of this work. One of the goals, for example, is to demonstrate that multiscale detection works and can be improved in the future.

- How does it compare in terms of deployment effort?

R: We added the following comments: the benefit is configuring remotely without being present in the environment to map the positions of fiducial markers, unlike if real markers are used, which require an extensive image database for training and testing. The disadvantage of implementing the tape-shaped marker is printing the segments and gluing them in an orderly manner in the environment in an aligned manner. The degradation of the marker due to humidity and exposure to sunlight was also noted in lines 634 to 636 in the Conclusions section.

 This is a crucial aspect to clarify and consider, especially for large spaces such as shopping malls and airports, which are mentioned as potential application scenarios. Do the benefits outweigh the deployment effort when compared, for example, with the use of Wi-Fi that uses already existing infrastructure?

R: Wi-Fi can have points of ambiguity, depending on the implementation, such as not distinguishing whether the person is close to a wall on the internal or external side. The fiducial marker does not generate this ambiguity because when the system recognizes a marker, this marker is physically in this place. Furthermore, Wi-Fi depends on electricity to function; if there is a lack of electricity, the location system stops working, although using the fiducial marker, it continues in operation as it does not need electricity to perform its function. We added this discussion to lines 537 to 547 in the Discussion section.

 Wi-Fi systems can cover larger areas with fewer access points, providing broader coverage with less physical deployment effort. Physically placing and aligning markers throughout the environment is time-consuming, especially in larger spaces. 

R: We agree that it is a manual and time-consuming task, depending on the coverage area size. We add this disadvantage at work. We added this to lines 537 to 540 in the Discussion section.

The coverage area of the markers may be limited, potentially requiring more markers for comprehensive coverage. For example, in large open areas (airports, shopping malls), the positioning performance will degrade or be impossible to estimate.

R: The tape-shaped marker can be encoded with 2^70 combinations that allow it to cover a large area, or even expand the number of child elements on the tape. Considering that a child element of the tape corresponds to 0.23 m in width, covering a perimeter of 1000 meters would require approximately 4348 different codes. This way, coding is flexible to cover large areas. We added this comment on lines 259 to 262 and 530 to 536. 

- What are the maintenance requirements or calibration procedures for the tape-shaped markers over time?

 R: The proposed marker detection system does not require calibration, and over time, only physical degradation of the markers occurs due to exposure to humidity and sunlight.

- The authors refer that the detection algorithm is robust against lighting variations, however, it does not address challenges related to other environmental factors such as shadows, reflections, or occlusions.

 R: New tests of robustness to lighting variations and robustness to difficult viewing angles simulated in a 3D environment have been added. However, the new tests do not solve the challenges of robustness to shadows and reflections. Added a paragraph on lines 402 to 424 in the Experiments and Results section.

A better discussion of these aspects is in my opinion essential to clarify which use cases this approach would be viable or better than other approaches.

R: These suggestions were added in the Discussion section on lines 537 to 547 to clarify for readers the use of the tape-shaped marker in a location system.

Reviewer 2 Report

Comments and Suggestions for Authors

MDPI Sensors Journal (Manuscript ID: sensors-3045106)

Comments to the Author

This paper proposes an indoor localization system using computer vision and employs tape-shaped markers using a smartphone. The authors are studying a useful research topic and proposing an interesting approach by testing it in two experiments, one virtual and one real-life. However, several points need to be addressed to improve the quality of the manuscript. Suggestions for improvement are provided below:

1.     Introduction: The introduction (and literature review) of existing localization technologies is generally kept very brief. It is worth elaborating on or at least providing proper references to relevant studies in this field. For example, in line 20, please provide established references for each localization technology mentioned in the paper with respect to the studies. Some examples include: for Bluetooth (https://doi.org/10.1016/j.buildenv.2020.106681), for Wi-Fi (https://doi.org/10.1016/j.enbuild.2018.06.040), and for Zigbee (10.1109/ICCSNT.2011.6182313). Additionally, among other technologies (line 21), hybrid approaches (i.e., sensor fusion: https://doi.org/10.1016/j.buildenv.2022.10968) are overlooked. There were many studies that combined various sensor data for indoor localization, which is worth mentioning to give a comprehensive overview to readers.

2.     Related Works: In lines 156-164, the authors explain the drawbacks and several issues that come with existing approaches, which are clearly presented. I would suggest further elaborating on how this study attempts to address those limitations. Include the contributions of this study to further elaborate on the novelty of this work.

3.     Methodology & Experiments: This section is written clearly and comprehensively. Sharing the code for reproducibility is a plus. I would like to understand more about the experiment comparisons between the 3D simulation and the real environment. As we know, the simulation environment is more controlled. Please elaborate on how this reflects real-world conditions. What are the limitations reflected in your experiments? Additionally, during mobile device testing, would the results be affected by using a different smartphone model or operating system? To what extent are these results generalizable?

Comments on the Quality of English Language

The quality of English is acceptable.

Author Response

Review 2

Comments to the Author

This paper proposes an indoor localization system using computer vision and employs tape-shaped markers using a smartphone. The authors are studying a useful research topic and proposing an interesting approach by testing it in two experiments, one virtual and one real-life. However, several points need to be addressed to improve the quality of the manuscript. Suggestions for improvement are provided below:

R: OK

  1.     Introduction: The introduction (and literature review) of existing localization technologies is generally kept very brief. It is worth elaborating on or at least providing proper references to relevant studies in this field. For example, in line 20, please provide established references for each localization technology mentioned in the paper with respect to the studies. Some examples include: for Bluetooth (https://doi.org/10.1016/j.buildenv.2020.106681), for Wi-Fi (https://doi.org/10.1016/j.enbuild.2018.06.040), and for Zigbee (10.1109/ICCSNT.2011.6182313). Additionally, among other technologies (line 21), hybrid approaches (i.e., sensor fusion: https://doi.org/10.1016/j.buildenv.2022.10968) are overlooked. There were many studies that combined various sensor data for indoor localization, which is worth mentioning to give a comprehensive overview to readers.

R: The title and abstract of the work were changed to inform the reader that the objective of this work is to present a proposal for a fiducial marker for indoor location systems.

The introduction section on lines 20 to 27 has been rewritten, adding some recommended citations.

 The references Bluetooth (https://doi.org/10.1016/j.buildenv.2020.106681),  Wi-Fi (https://doi.org/10.1016/j.enbuild.2018.06.040), and hybrid approaches (i.e., sensor fusion: https://doi.org/10.1016/j.buildenv.2022.10968) were not found, but other references were added to these technologies mentioned in the Introduction section.

  1.     Related Works: In lines 156-164, the authors explain the drawbacks and several issues that come with existing approaches, which are clearly presented. I would suggest further elaborating on how this study attempts to address those limitations. Include the contributions of this study to further elaborate on the novelty of this work.

R: New contributions from this work were detailed in the paragraph above on lines 165 to 175. They mention that the markers in Table 1 do not clearly inform the number of different codes the marker has. The present work proposes more than 1 million different coding combinations, which have coding based on the level of black and white ink with positive or negative values. The markers mentioned in Table 1 lack an interface for generating markers for large areas. This work allows the generation of the marker using drawings on the floor plan of room grids. It can also be enlarged or reduced according to the need for use with a 25:7 aspect ratio to adapt to the environment visually.

  1.     Methodology & Experiments: This section is written clearly and comprehensively. Sharing the code for reproducibility is a plus. I would like to understand more about the experiment comparisons between the 3D simulation and the real environment. As we know, the simulation environment is more controlled. Please elaborate on how this reflects real-world conditions. What are the limitations reflected in your experiments? Additionally, during mobile device testing, would the results be affected by using a different smartphone model or operating system? To what extent are these results generalizable?

R: New tests were added to the simulations in a 3D environment that made it possible to evaluate robustness to lighting variations and robustness to difficult viewing angles. We added this in lines 402 to 424. In addition to the existing tests, in controllable environments, to identify the limitations of the detection algorithm. Unlike tests in a real environment where it was difficult to control the variation in lighting, shadows, and camera blur in the localization test. However, it was possible in both environments to verify the efficiency of detection on multiple scales, which is the main characteristic of this marker system. We added this discussion in lines 600 to 615 of the Discussion section. The results in the real test can differ when using a different mobile device, especially when using lower processing and camera configurations than those used in the test. However, we inform the readers of the device configurations to make possible a comparison.

Round 2

Reviewer 2 Report

Comments and Suggestions for Authors

Thank you for addressing my comments on related works, methodology, and experiments. 

Regarding the Introduction, the authors mentioned that they have not been able to find the suggested readings for respective localization technology using DOIs. Here is the breakdown of the study names and respective DOIs to be included, to further enrich the introduction.

  1. Bluetooth: A scalable Bluetooth Low Energy approach to identify occupancy patterns and profiles in office spaces (2020) (https://doi.org/10.1016/j.buildenv.2020.106681)
  2. WiFi: Device-free occupancy detection and crowd counting in smart buildings with WiFi-enabled IoT (2018) (https://doi.org/10.1016/j.enbuild.2018.06.040)
  3. Hybrid Approaches: Occupancy prediction using deep learning approaches across multiple space types: A minimum sensing strategy (2022) (https://doi.org/10.1016/j.buildenv.2022.109689)
  4. ZigBee: Suggested study has been found.
Comments on the Quality of English Language

Quality of English is satisfactory